# FINE-TUNING WITH VERY LARGE DROPOUT

## ABSTRACT

It is impossible today to pretend that the practice of machine learning is compatible with the idea that training and testing data follow the same distribution. Several authors have recently used ensemble techniques to show how scenarios involving multiple data distributions are best served by representations that are both richer than those obtained by regularizing for the best in-distribution performance, and richer than those obtained under the influence of the implicit sparsity bias of common stochastic gradient procedures.

This contribution investigates the use of very high dropout rates instead of ensembles to obtain such rich representations. Although training a deep network from scratch using such dropout rates is virtually impossible, fine-tuning a large pre-trained model under such conditions is not only possible but also achieves out-of-distribution performances that exceed those of both ensembles and weight averaging methods such as model soups.

This result has practical significance because the importance of the fine-tuning scenario has considerably grown in recent years. This result also provides interesting insights on the nature of rich representations and on the intrinsically linear nature of fine-tuning a large network using a comparatively small dataset.

## 1 INTRODUCTION

The practice of machine learning has been shaped by the assumption that training and testing examples are independently drawn from the same unknown probability distribution. *This is seldom the case in modern settings, not only because this i.i.d. assumption breaks down for the problems of interest, but also because it is often convenient to use multiple datasets that are known to follow different distributions.* For instance, we may pre-train a deep network on a large dataset, fine-tune it on a smaller dataset specific to the task of interest, and test on a collection of tasks designed to benchmark various aspects of the system.

Many of the tenets of machine learning should therefore be regarded with healthy suspicion. For instance, under the i.i.d. assumption, favoring solutions with sparse representations has well-known benefits on the generalization performance. Yet, several authors (Zhang et al., 2022; Zhang & Bottou, 2023; Chen et al., 2023) make the point that scenarios involving multiple distributions are best served by "*richer representations*" that contain redundant features, that is, features that do not improve the model performance on the training distribution, but could prove helpful when the distribution changes.

It would be nice to construct such rich representations by merely optimizing the expectation of a suitable loss function for a single training distribution, for instance using stochastic gradient techniques. Alas, this hope is contradicted by the implicit sparsity bias of stochastic gradient algorithms (Andriushchenko et al., 2023; Blanc et al., 2020). In a nutshell, a feature only survives when it brings an incremental training error advantage relative to what can be achieved using all the other features already present in the network. We slightly abuse the terminology and call them "strongly relevant". However, features that are not strongly relevant might nevertheless

(a) be incrementally useful when the data follows a different distributions of interest, or
(b) be useful under the training distribution when added to certain subsets of the other existing features instead of all of them ("weakly relevant").

It is therefore tempting to "enrich" the representation with features of type (b), which can be found using the training data, and hope that some of these will turn out to also be features of type (a) whose inclusion helps when the data distribution changes.

The dropout technique (Srivastava et al., 2014) seems well suited to finding weakly relevant features because randomly masking units of a representation layer during training amounts to forming random subsets of all other available features. However, in order to form small subsets, one would have to use very high levels of dropout. Unfortunately, training a sizable deep network from scratch with such a large dropout is practically impossible. Instead, computationally demanding methods, such as adversarial sampling (Zhang et al., 2022; Chen et al., 2023) and representation ensembles (Zhang & Bottou, 2023), have been proposed to find weakly relevant features while training a network from scratch.

There is however a practically meaningful scenario in which we can use an extremely aggressive dropout: fine-tuning a pre-trained network using a comparatively small dataset. This is possible because such a fine-tuning operation makes only modest changes to the network weights. For example, several authors (Ramé et al., 2022b; Wortsman et al., 2022a) argue that fine-tuned networks remain "*linearly connected*", that is averaging the parameters of multiple fine-tuned networks approximate the ensemble of these networks. Evci et al. (2022) even show that a linear classifier on top of the union of internal-layer features of pre-trained residual networks can match or exceed the performance of fine-tuning.

In the present work, we adopt the out-of-distribution fine-tuning setup (*three-distributions*) of Ramé et al. (2022b). In this framework, we have access to a model pre-trained using a large dataset for a task weakly related to the task of interest. This pre-trained model is then fine-tuned on datasets that illustrate the task of interest, and then tested on a dataset for the same task but with a different distribution. However, instead of enriching the representations by constructing ensembles (Zhang & Bottou, 2023) or averaging weights (Ramé et al., 2022b;a; Wortsman et al., 2022b), we simply *fine-tune using very large dropout levels*, randomly masking above 90% of the units in the representation layer. We find that this simple approach *exceeds the performance of both ensemble and weight-averaging methods*. This result is not only *practically meaningful*, but also clarifies the idea of *richer representation*.

## 2 RELATED WORK

**Constructing versatile representations**   Reusing or transferring features across related tasks has been commonplace for more than one decade (Collobert et al., 2011; Bottou, 2011; Sharif Razavian et al., 2014) and plays a fundamental role in the appeal of foundational models (Bommasani et al., 2021a). However, once the optimization process has identified a set of features that is sufficient to achieve near-optimal performance on the training set, additional features are often discarded because they do not bring an incremental benefit to the training error, despite the fact that they may independently carry useful information (Zhang & Bottou, 2023).

Researchers have devised ways to obtain more versatile representations by engineering a diversity of datasets, architectures, and even hyper-parameters (Chen et al., 2020; Wang et al., 2022; Dvornik et al., 2020; Bilen & Vedaldi, 2017; Gontijo-Lopes et al., 2021; Li et al., 2021; 2022; Chowdhury et al., 2021), as an alternative to the most popular approach which consists of simply using ever larger datasets (Bommasani et al., 2021b).

Interesting results have also been obtained without engineering diversity and without increasing the dataset sizes. Zhang et al. (2022) and Chen et al. (2023) propose to discover rich representation through multiple training episodes that adversarially reweigh the training dataset to impede the use of previously learned features. Zhang & Bottou (2023) show that surprisingly good results can be obtained by concatenating the representations of multiple networks that are trained in exactly the same way, save for the random seed used in the stochastic gradient process.

**Fine-tuning as a near-linear process**   Although modern deep residual networks feature highly complex nonconvex cost functions, several authors have shown that their final training phase remains mostly confined to a nearly-convex attraction basin (Izmailov et al., 2018; Li et al., 2018; Frankle et al., 2020). The same observation holds when fine-tuning a large pre-trained network using a dataset

whose size is considerably smaller than the dataset size one would need to train such a large network from scratch. As long as one starts from the same pre-trained model, Wortsman et al. (2022a) and Ramé et al. (2022b;a) observe that averaging the weights of diverse fine-tuned models can reproduce the i.i.d. and o.o.d. performances of the ensemble of these models, implying that fine-tuning is a near-linear process.

Maddox et al. (2021) and Mu et al. (2019) propose instead to approximate the fine-tuning process with a first-order Taylor expansion, obtaining a linear system operating on top of the NTK features. Evci et al. (2022) match the performance of fine-tuning by merely learning a strongly regularized linear model that takes all internal layer states as inputs. Meanwhile, Yu et al. (2023) efficiently fine-tune large foundational language models by essentially restricting the weight updates to low dimensional manifolds.

**Fine-tuning with very large dropout**  Our contribution advocates using very large dropout in the fine-tuning scenario in order to force the learning algorithm to create a redundant representation without specifically engineering diversity. We do not seek to propose new dropout variations (Chu et al., 2022), understand dropout from either an overfitting/underfitting perspective (Liu et al., 2023) or from a Bayesian perspective (Gal & Ghahramani, 2016).

## 3 FINE-TUNING AND DROPOUT

### 3.1 THE THREE-DISTRIBUTIONS SETUP

The ***two-distributions*** setup is a commonly used for transfer learning. In this setup, features $\Psi$ are obtained by pre-training a network on a large training set associated with a first distribution $\mathcal{T}_\mathrm{p}$. These features are then used to construct or initialize a new model $\omega_\mathrm{d} \circ \Psi$, which is then trained using a smaller training set associated with a second distribution $\mathcal{T}_\mathrm{d}$. The question is to determine which pre-training approach is most likely to make the features $\Psi$ useful for the transfer task $\mathcal{T}_\mathrm{d}$.

The ***three-distributions*** setup (Ramé et al., 2022b) views the pre-trained model as a base model that is assumed very rich but whose training process is beyond our control (e.g., a fundational model). The features $\Psi$ of the pre-trained model are then incorporated into a new model $\omega_\mathrm{d} \circ \Psi$ that is fine-tuned using a second distribution $\mathcal{T}_\mathrm{d}$ and eventually tested on a third distribution $\tilde{\mathcal{T}}_\mathrm{d}$ illustrating the same general task as the second distribution (e.g., using the same classification labels.) The question is then to determine which fine-tuning approach is most likely to produce a model that will perform robustly under the eventual testing distribution $\tilde{\mathcal{T}}_\mathrm{d}$.

### 3.2 METHOD

The key results described later in this paper have been obtained with a very simple method. The base model is a deep learning network with residual connections trained on data $\mathcal{T}_\mathrm{p}$ that is related to but substantially larger than the datasets illustrating the task of interest. Some of these datasets ($\mathcal{T}_d$) are used to fine-tune the base model. Performance is reported on both held-out data from the fine-tuning datasets (i.i.d. performance on $\mathcal{T}_d$) and data from the remaining datasets (o.o.d. performance on $\tilde{\mathcal{T}}_d$).

We focus on residual networks because fine-tuning has been found to hardly change the inner layers of non-residual networks (Raghu et al., 2019, fig 2). In contrast, skip connections in residual networks expose the inner block features in such a manner that the fine-tuning process can utilize these features in a near-linear way (Evci et al., 2022).

Fine-tuning is carried out with a standard stochastic learning procedure (e.g. SGD or ADAM) after applying a very large dropout to the penultimate layer representation $\Phi$. Unlike (Srivastava et al., 2014), we only apply dropout on the penultimate layer representation $\Phi$,[1] because skip connections in residual networks expose many inner-layer features to the last linear layer, as illustrated by the

---

[1]Except in Figure 6 in the appendix where we investigate the effect of dropping entire inner residual blocks during fine-tuning.

decomposition of residual networks proposed by Veit et al. (2016),

$$\Phi(x) = \underbrace{x}_{\phi_0(x)} + \underbrace{f_1(x)}_{\phi_1(x)} + \underbrace{f_2(x + f_1(x))}_{\phi_2(x)} + \cdots = \sum_{i \in [0,\dots,l]} \phi_i(x) \in \mathbb{R}^p \,, \tag{1}$$

where $f_i$ represents the function implemented by the $i$-th residual block, and

$$\Phi_{\text{dropout}}(x) = \frac{m(\lambda)}{1-\lambda} \odot \Phi(x) \,, \tag{2}$$

where $\odot$ represents the component-wise product and $m(\lambda)$ is a vector of random Bernoulli variables equal to 0 with probability $\lambda$ and 1 with probability $1 - \lambda$. The additive decomposition of $\Phi(x)$ in equation equation 1 makes clear that applying dropout to $\Phi(x)$ simultaneously blocks the contributions $\phi_i(x)$ of all residual blocks.

## 4 EXPERIMENTS

We perform most experiments using PACS (Li et al., 2017), VLCS (Fang et al., 2013), OFFICE HOME (Venkateswara et al., 2017), and TERRA INCOGNITA (Beery et al., 2018) datasets. These datasets are part of the DOMAINBED suite which has been widely used for this kind of experiments.[2] With $9,991$ to $24,788$ examples, they are substantially smaller than the pre-training dataset IMAGENET with 1.2M examples. We also use the larger DOMAINNET dataset (Peng et al., 2019), 0.58M examples, to show that linear connectivity breaks down when the fine-tuning dataset size becomes comparable with the pre-training dataset size and justifies carrying out many more fine-tuning iterations.

Each of these datasets is divided into four sub-datasets that share the same target label categories but follow a different distribution. For example, one sub-dataset of PACS contains simple sketch images of 'dog' and 'elephant', while another sub-dataset contains real photos of 'dog' and 'elephant'. This makes it possible to conveniently evaluate o.o.d. performance by fine-tuning on three sub-datasets and testing on the fourth one.

Fine-tuning experiments are carried out on two widely used residual architectures: convolutional residual networks, RESNETs (He et al., 2016), and vision transformers, VITs (Dosovitskiy et al., 2020). Unless otherwise stated, all convolutional residual network experiments are carried out using the RESNET50 neural network pre-trained on IMAGENET with substantial data augmentations[3] such as TRIVIALAUGMENT (Müller & Hutter, 2021), CUTMIX (Yun et al., 2019), and RANDOM ERASINGS (Zhong et al., 2020). These augmentations mimic the properties of large foundational models that learn substantial diverse features using very large and diverse pre-training data. We refer to this network as RESNET50 #2 as opposed to the original RESNET50 #1 recipe described by He et al. (2016). Similarly, all visual transformer fine-tuning experiments leverage a IMAGENET pre-trained VIT-L-16 model[4] with 304M parameters.

### 4.1 VERY LARGE DROPOUT YIELDS BETTER O.O.D. PERFORMANCE

Using these same datasets, Gulrajani & Lopez-Paz (2020) argue that plain Empirical Risk Minimization (ERM) equals and often betters the o.o.d. performance of purposefully designed methods, such as CORAL (Sun & Saenko, 2016), VREX (Krueger et al., 2021), and IRM (Arjovsky et al., 2019). More recently, Arpit et al. (2022), Cha et al. (2021), Ramé et al. (2022b), and Ramé et al. (2022a) find that ensemble and weight averaging methods consistently outperform the o.o.d. performance of ERM. We now show that fine-tuning with very large dropout outperforms the o.o.d. performance of these state-of-the-art methods.[5]

Following these earlier works, we focus on the o.o.d. performance of these methods because this is the testing performance that matters for the practical situations that the three-distribution setup

---

[2]https://github.com/facebookresearch/DomainBed
[3]https://pytorch.org/blog/how-to-train-state-of-the-art-models-using-torchvision-latest-primitives/
[4]https://github.com/pytorch/vision/tree/main/references/classification#vit_l_16
[5]Code: https://anonymous.4open.science/r/verylarge_dropout-2BCB/

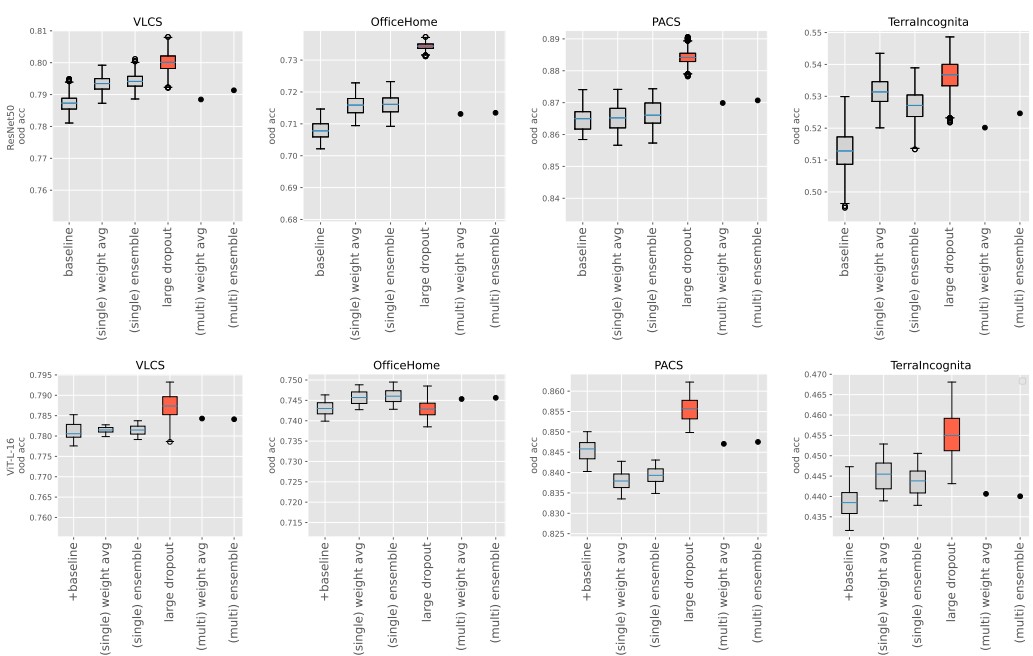

Figure 1: o.o.d. performance comparison between very large dropout, ensembles, and weight averaging methods on four DOMAINBED tasks and two backbones (RESNET50 and VIT-L-16). Baseline results were obtained using plain fine-tuning with different hyperparameters. **Weight averaging** results either average the model weights collected every 300 iterations along each fine-tuning trajectory or the final model weights of all fine-tuning trajectories as in (Ramé et al., 2022b). **Ensemble** results average instead the model outputs. Finally, **large dropout** results were obtained like the baseline results but using a 90% dropout rate on the penultimate layer. Each box summarizes the results obtained with 1296 hyper-parameters combinations (no hyper-parameter selection).

Table 1: o.o.d. performance comparison between very large dropout, ensembles, and weight averaging methods after hyperparameter selection. The hyperparameter is selected according to the best i.i.d. performance.

| | dataset | baseline | weight avg (single) | ensemble (single) | large dropout | weight avg (multi) | ensemble (multi) |
|---|---|---|---|---|---|---|---|
| ResNet | VLCS | 78.3 | 79.4 | 79.6 | **80.1** | 78.8 | 79.1 |
| | OFFICE HOME | 71.4 | 72.2 | 72.3 | **73.6** | 71.3 | 71.3 |
| | PACS | 87.3 | 86.9 | 87.3 | **88.5** | 87.0 | 87.1 |
| | TERRA INCOGNITA | 51.0 | 53.1 | 52.3 | **53.9** | 52.0 | 52.5 |
| | **Average** | 72.0 | 72.9 | 72.9 | **74.0** | 72.3 | 72.5 |
| ViT-L-16 | VLCS | 78.1 | 78.1 | 77.9 | **79.0** | 78.4 | 78.4 |
| | OFFICE HOME | 74.6 | **74.8** | **74.8** | 74.6 | 74.5 | 74.6 |
| | PACS | 85.0 | 84.2 | 84.3 | **86.0** | 84.7 | 84.8 |
| | TERRA INCOGNITA | 44.4 | 45.1 | 44.8 | **45.8** | 44.1 | 44.0 |
| | **Average** | 70.5 | 70.6 | 70.5 | **71.4** | 70.4 | 70.5 |

targets. The i.i.d. performance must then be viewed as a secondary indicator, akin to the training set performance, which could be used to cast a light on the training process but cannot be used as a predictor of the system performance under its actual testing distribution.

Figure 1 reports the o.o.d. performances of the following approaches.

- **Baseline** results are obtained by fine-tuning our RESNET50 or VIT-L-16 using SGD with 0.9 momentum for $10,000$ iterations.[6] A 10% learning rate decay is applied at $5000^{th}$ iterations. For each choice of three training sub-datasets, we repeat three experiments for each combination of learning rate in $\{10^{-3}, 5.10^{-4}\}$ and L2 weight decay in $\{10^{-4}, 5.10^{-5}, 10^{-5}\}$. We measure the i.i.d. performance using 20% examples held from the training data, and we measure o.o.d. performance on the fourth sub-dataset. Each box summarizes the performances obtained using two learning rate choices and three weight decay choices, for four possible training dataset choices, resulting in $(2 \times 3)^4 = 1296$ results. Following Gulrajani & Lopez-Paz (2020), we prevent overfitting by early-stopping at the best i.i.d. performance.

- **Dropout** results are obtained using the same protocol but using a 90% dropout rate on the penultimate layer representation.

- **Ensemble** results are obtained in two ways, either using an ensemble of checkpoints collected along each fine-tuning trajectory, or using the ensemble of the final checkpoints collected along all fine-tuning trajectories with different hyper-parameters.

- **Weight averaging** results approximate the corresponding ensembling results by averaging the model weights instead of averaging the model outputs.

As expected, both ensemble methods (Ueda & Nakano, 1996; Dietterich, 2000) and their weight averaging approximation (Ramé et al., 2022b; Wortsman et al., 2022a) improve on the o.o.d. baseline performance. However, fine-tuning with a very large dropout outperforms the o.o.d. performance of both ensemble and weight averaging methods. There is even a *large gap* between the worst dropout results and the best ensemble results for the OFFICE HOME and PACS datasets. Figure 7 in the Appendix shows that the i.i.d. performance of the large dropout method lags behind that of ensembles, revealing that the o.o.d. performance improvement is not a secondary effect of some i.i.d. performance improvement (i.e. the o.o.d. performance gaps in o.o.d. Figure 1 do not come from i.i.d. overfitting/underfitting.).

The box plots in Figure 1 summarize the o.o.d. performance of all possible hyper-parameter combinations. Following again Gulrajani & Lopez-Paz (2020), Table 1 features o.o.d. performances obtained using the hyper-parameter combination that provides the best i.i.d. performance. Because RESNET50 produces a better performance than VIT-L-16 on these o.o.d. fine-tuning tasks, our experiments in the following sections will be conducted on RESNET50.

### 4.1.1 OPTIMAL O.O.D. DROPOUT RATE

To the best of our knowledge, such large dropout rates (90% and above) are considered unsuitable for training a network from scratch and have not been previously used for fine-tuning either. This section illustrates how the optimal dropout rate can be very high in fine-tuning scenarios and falls to small values when one gets closer to training the network from scratch.

Figure 2 compares various dropout rates on the four DOMAINBED tasks. A 90% dropout rate reliably produces good o.o.d. performance on all four tasks. The optimal dropout rate for o.o.d. performance ranges from 90% to 95% for VLCS and PACS task (with 10k examples). And becomes slightly smaller, about 90%, for the slighlty larger datasets OFFICE HOME and TERRA INCOGNITA (with 15k to 25k examples).

**Increasing the fine-tuning dataset size to approach the pre-training dataset size** The larger the fine-tuning dataset, the more fine-tuning iterations we can make without overfitting. When the fine-tuning dataset size approaches the pre-training dataset size, the difference between fine-tuning and training from scratch becomes less clear, *the linear connectivity property disappears, the linear approximation perspective on fine-tuning no longer holds, and the optimal dropout rate falls sharply*.

---

[6]We use a batch size 32 for all RESNET fine-tunings, and reduce the batch size to 16 for all VIT-L-16 fine-tunings due to the VRAM constraint.

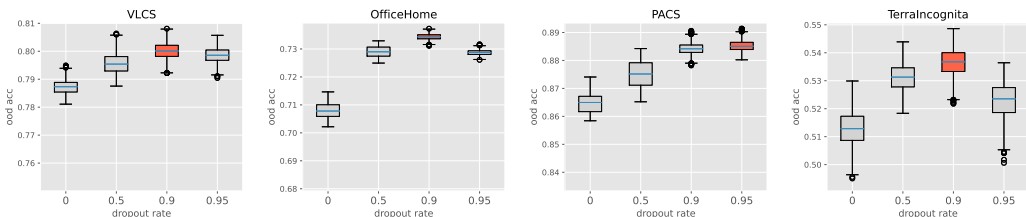

Figure 2: Effect of diverse dropout rates during fine-tuning. The best o.o.d. performances are attained using rates around or above 90%. A large dropout rate (e.g. 90%) reliably produces good o.o.d. performance on all four tasks.

Figure 3 illustrates this effect using the larger DOMAINNET dataset (Peng et al., 2019) that contains 586k examples (almost half as big as IMAGENET) and requires 30, 000 fine-tuning iterations.

**Training from scratch on the fine-tuning dataset** Figure 4 shows the effect of various dropout rates when one trains a network on the VLCS task from scratch, that is starting from a randomly initialized network without pretraining (i.e. the pretraining dataset size is zero). The optimal dropout rate falls to about zero. Dropout rates higher than 50% have a negative impact on both the i.i.d. and the o.o.d. performance of the network. *This suggests again that high dropout rates make it difficult to create new features (a nonlinear operation), but does not prevent leveraging existing features that were possibly buried in the network inner layers (a linear operation).* This is the idea of richer representation we discussed in section 1. We will provide other fine-tuning experiments on the richer representation idea in Section 4.3.

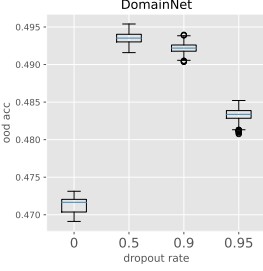 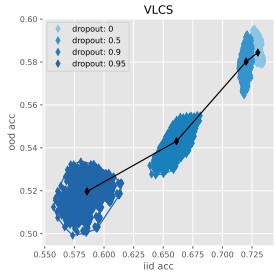

Figure 3: Comparison of various dropout rates on the larger DOMAINNET dataset (586K examples), whose size approaches the pretraining dataset size (IMAGENET, 1.2M examples). The optimal dropout rate falls to about 50%, a value comparable to the dropout rates traditionally used when training from scratch.

Figure 4: Comparison of dropout rates when training a RESNET50 network *from scratch* on the VLCS dataset. The optimal dropout rate falls to about zero. Dropout rates greater than 50% negatively impact both the i.i.d. and the o.o.d. performances.

### 4.2  POPULAR FINE-TUNING TECHNIQUES DO NOT SUBSTANTIALLY IMPROVE THE O.O.D. PERFORMANCE OF LARGE DROPOUTS

Various fine-tuning techniques have been proposed to improve the fine-tuning ability to leverage the representations learned by a pre-trained model, such as using a larger learning rate on the last layer (Caron et al., 2020; Bardes et al., 2021; Kumar et al., 2022) or, as discussed above, using weight averaging and ensemble methods (Ramé et al., 2022b;a; Arpit et al., 2022). We show in this section that using these techniques *in addition to very large dropout rates* do not yield much o.o.d. performance improvements over using large dropout rates alone. Note that we still expect some incremental benefits because both weight averaging and ensembles reduce the stochastic optimization noise and accelerate training in general (Polyak & Juditsky, 1992).

| dataset | baseline | baseline + 10× last-layer lr | baseline + large dropout | baseline + large dropout + 10× last-layer lr |
|---|---|---|---|---|
| VLCS | 78.3 | 79.9 (+1.6) | 80.1 (+1.8) | **80.5** (+2.2) |
| OFFICE HOME | 71.4 | 71.8 (+0.4) | **73.6** (+2.2) | 73.3 (+1.9) |
| PACS | 87.3 | 87.0 (-0.3) | **88.5** (+1.2) | 88.3 (+1.0) |
| TERRA INCOGNITA | 51.0 | 52.2 (+1.2) | 53.9 (+2.9) | **54.9** (+3.9) |
| Average | 72.00 | 72.73 | 74.03 | 74.25 |

Table 2: Incremental benefits achieved by applying a $10\times$ larger learning rate in the last layer. The first two columns show that this $10\times$ last-layer learning rate is helpful to baseline. Then the middle two columns show that using a large dropout rate vastly improves the o.o.d. performance of merely using the increased learning rate ($\sim 1.3\%$). The last two columns reveals that using this $10\times$ larger last-layer training rate yields small or zero incremental improvements over only using a large dropout rate ($\sim 0.2\%$).

### 4.2.1 LARGE LEARNING RATES FOR THE LAST LAYER

Several authors routinely use a larger training rate on the last layer on the intuition that fine-tuning a pre-trained deep network on a different target task entails training a new last layer from scratch (Caron et al., 2020; Bardes et al., 2021; Kumar et al., 2022).

Table 2 follows a similar fine-tuning process as in Table 1 but uses a $10\times$ larger training rate for the last layer classifier. Comparing the last two columns in Table 2 shows that using this $10\times$ larger last layer training rate yields small or zero incremental improvements over only using a large dropout rate ($\sim 0.2\%$). Comparing the middle two columns further shows that using a large dropout rate vastly improves the o.o.d. performance of merely using the increased learning rate ($\sim 1.3\%$).

### 4.2.2 ENSEMBLE AND WEIGHT AVERAGING

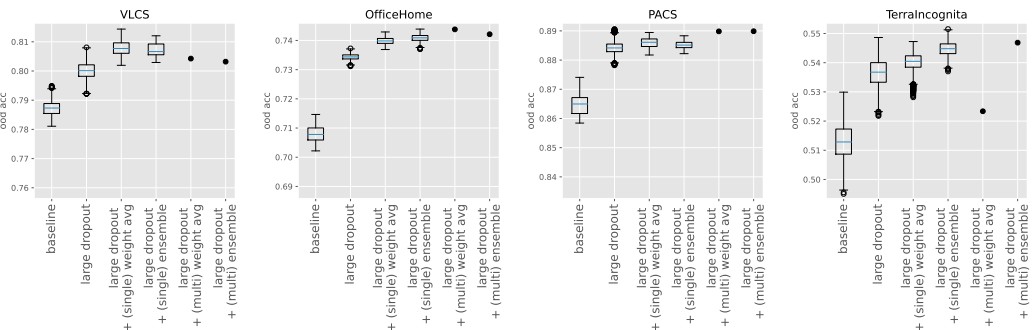

Figure 5: Incremental benefits achieved by constructing ensembles or by averaging the weights of models fine-tuned with very large dropouts. The baseline and dropout results are the same as those reported in Figure 7. In contrast, the ensemble and weight averaging results are now obtained by averaging the output or the weights of models fine-tuned *with large dropouts*. Ensemble and weight averaging techniques provide a marginal o.o.d. performance improvement on VLCS or OFFICE HOME and a negligible o.o.d. performance improvement on PACS or TERRA INCOGNITA.

Figure 5 similarly explores the incremental benefits achieved by constructing ensembles or by averaging the weights of models fine-tuned with very large dropouts. The incremental improvements in o.o.d. performance achieved by these methods, if any, are much smaller than the improvement achieved by large dropout rates alone. Comparing Figures 7 and 5 also shows that in contrast, fine-tuning with large dropout rates before computing ensembles or averaging model weights brings large o.o.d. performance improvements over fine-tuning without dropout.

| dataset | RESNET #1 naive fine-tune | RESNET #1 advance fine-tune | RESNET #2 naive fine-tune | RESNET #2 advance fine-tune |
|---|---|---|---|---|
| VLCS | 76.7 | 79.5 | 78.3 | 81.0 |
| OFFICE HOME | 68.9 | 71.8 | 71.4 | 74.3 |
| PACS | 86.2 | 87.5 | 87.3 | 89.2 |
| TERRA INCOGNITA | 48.2 | 49.7 | 51.0 | 55.8 |
| **Average** | 69.0 | 72.1 | 72.0 | 75.1 |

Table 3: Comparison of the o.o.d. performances obtained after fine-tuning two pre-trained networks: RESNET #1 and RESNET #2. Hyperparameters are selected according to the best i.i.d. performance. Compared with RESNET #1 (He et al., 2016), RESNET #2 was pre-trained with the vast array of data augmentation techniques. For each of these two pre-trained networks, we follow two fine-tuning approaches: 1) naive fine-tuning; 2) advanced fine-tuning including various tricks intended to improve the o.o.d. performance, e.g. large dropout (90%), weight averaging, and increased last-layer learning rate, using hyper-parameters are selected according to the i.i.d. performance. Despite all this technology, advanced fine-tuning of a pretrained RESNET #1 (2nd column) barely matches the performance of naive fine-tuning on RESNET #2 (3rd column).

### 4.3 RICHER PRE-TRAINING BEATS SOPHISTICATED FINE-TUNING

We have demonstrated that the very-large dropout method delivers consistently better o.o.d. performance than computing ensembles or weight-averages of models fine-tuned without dropout. However we also have argued that fine-tuning does not create new representations but merely exploits the representations already present in the pre-trained model. Therefore the final o.o.d. performance of this fine-tuning process must strongly depend on the quality and the diversity of the features present in the pre-trained network (*richer representation*), even if these features are not exploited by the pre-trained network but buried in its hidden layers.

To validate this assertion, we compare the i.i.d. and o.o.d. performance obtained by various methods applied to RESNET50 networks pre-trained using the same IMAGENET data but using different data augmentation schemes. As explained in the first paragraphs of section 4, the results reported so far use a network pre-trained using a broad array of data augmentation techniques, termed RESNET #2. We now compare its fine-tuning properties with network termed RESNET #1 pre-trained using the simpler protocol described in He et al. (2016).

Figure 3 compares the o.o.d. performances of both networks after regular fine-tuning and after fine-tuning with all the available tricks, that is, with dropout, with $10\times$ larger last layer learning rate, and after averaging the weights of checkpoints collected along the fine-tuning trajectory. This comparison makes clear that the quality of the pre-trained representation matters more than the sophistication of the fine-tuning techniques. This is consistent with the idea that fine-tuning only leverages the existing features of the pre-trained network and does not create new ones.

## 5 DISCUSSION

The o.o.d. performance of fine-tuning with very large dropout consistently exceeds that achieved by popular techniques such as ensemble and by more recent techniques such as weight averaging. Furthermore, ensemble and weight averaging techniques only bring a small incremental improvement when applied on top of fine-tuning with large dropout rates. This suggests that very large dropout implements a key factor that favors o.o.d. performance, which we believe is related to seeking features of type (a) among features of type (b) as explained in the introduction.

Both ensemble and weight-averaging techniques can be used for training a network from scratch or for fine-tuning a pre-trained network. In contrast, very large dropout rates cannot be realistically used when training a network from scratch. We argue that they work for fine-tuning because fine-tuning is well approximated as a linear process that can leverage their existing or buried features of a pre-trained network but cannot create new ones. Using large dropout rates is akin to a form of L2 regularization, expressing a richer set of features even if redundant.

This result also illustrates how the i.i.d. and o.o.d. scenarios can call for very different techniques. It is well known that sparse representations can be very helpful in the i.i.d. scenario, and it is increasingly clear that rich representations are preferable in the o.o.d. scenario (Zhang et al., 2022; Zhang & Bottou, 2023; Chen et al., 2023). There are no reasons to expect that the many techniques designed for the i.i.d. scenarios will systematically help o.o.d. generalization. The very-large dropout case is one of many such examples.

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

# Fine-tuning with Very Large Dropout

## Supplementary Material

## A    EXPERIMENT DETAILS

### A.1    FINE-TUNING DOMAINNET IN FIGURE 3

The DOMAINNET o.o.d. fine-tuning experiment in Figure 3 follows the same pipeline as other o.o.d. fine-tuning experiments on VLCS, PACS, OFFICE HOME, and TERRA INCOGNITA datasets. Due to the larger size of DOMAINNET dataset, we use larger learning rates $\{3.10^{-3}, 5.10^{-3}\}$ and a longer training iteration $30,000$ with a $10\%$ learning rate decay at $15,000$ and $25,000$.

### A.2    TRAINING FROM SCRATCH IN FIGURE 4

The VLCS scratch training experiment in Figure 4 follows the same pipeline as o.o.d. fine-tuning experiments. But it uses larger learning rates $\{5.10^{-3}, 10^{-2}\}$ on a random initialized RESNET50 network (all weights are trainable).

### A.3    COMPUTE RESOURCES

All experiments are done on V100 GPUs with Intel(R) Xeon(R) Gold 6230 CPUs. One V100 GPU and less than 32GB RAM are enough to fine-tune one Domainbed dataset within a few hours.

## B    ADDITIONAL RESULTS

### B.1    RESIDUAL BLOCK DROPOUT (STOCHASTIC DEPTH) IN O.O.D. FINE-TUNING

Huang et al. (2016) shows dropping a residual block at random with probability $p$ helps train very deep residual neural networks from scratch. Dropping a residual block, for example $f_1(x)$, in $\Phi(x)$ blinds the feature $\phi_1(x)$ but also changes the input of successive layers:

$$\Phi'(x) = \underbrace{x}_{\phi_0(x)} + \underbrace{f_1(x)}_{\phi_1(x)} + \underbrace{f_2(x + f_1(x))}_{\phi_2'(x)} + \dots$$

So an aggressive residual block dropout heavily disturbs the input of high residual blocks, hurts the fine-tuning process. Figure 6 showcase the effect of redisual block dropout in o.o.d. fine-tuning.

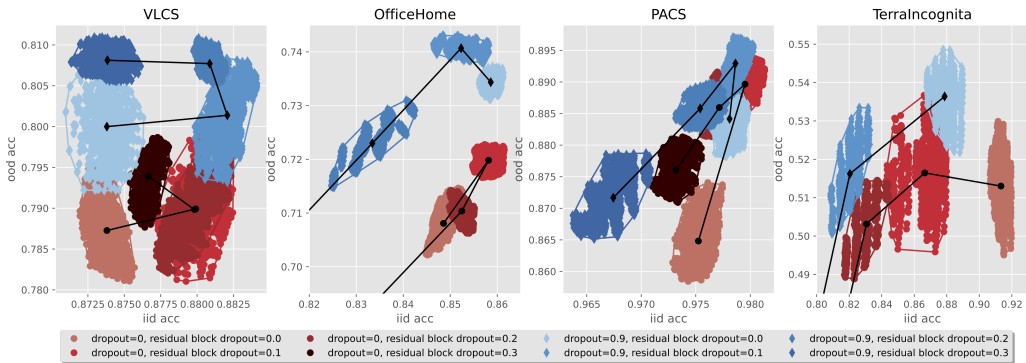

Figure 6: Residual block dropout (stochastic depth) and penultimate layer representation dropout comparison. **Residual block dropout** random drops a residual block with probability $\in \{0.1, 0.2, 0.3\}$. Mild **residual block dropouts** (0.1) provides some o.o.d. improvements (●), but lags behind **dropout** (◆) on VLCS, OFFICE HOME, and TERRA INCOGNITA. Aggressive residual block dropouts hurt finetuning (e.g. $\geq 0.3$ on OFFICE HOME and TERRA INCOGNITA), because dropping a residual block change the input of successive layers. Likewise the comparison of ensemble, weight averaging, and large last-layer learning rate in section 4.2, a proper residual block dropout helps fine-tuning in general, but is still a secondary factor compared with dropout.

## B.2    O.O.D. PERFORMANCE IMPROVEMENT IS NOT A SECONDARY EFFECT OF SOME I.I.D. PERFORMANCE IMPROVEMENT

Figure 7 provides both i.i.d. and o.o.d. performance of very large dropout, ensembles and weight averaging methods. The very large dropout method outperforms other methods on o.o.d. performance without any advantage on i.i.d. performance. It reveals that the o.o.d. performance gaps of different methods in Figure 7 and Figure 1 do not come from the i.i.d. overfitting/underfitting.

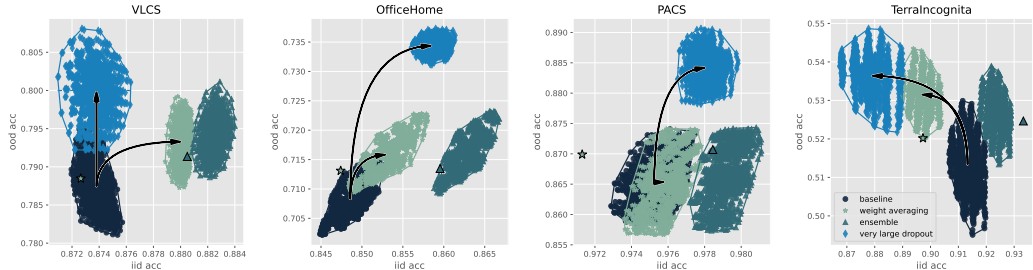

Figure 7: Performance comparison between very large dropout, ensembles, and weight averaging methods on four DOMAINBED tasks and RESNET50 #2 backbone. The horizontal axis denotes the i.i.d. performance and the vertical axis the o.o.d. performance. Baseline results were obtained using plain fine-tuning with different hyperparameters ($1296\times$●). **Weight averaging** results either average the model weights collected every 300 iterations along each fine-tuning trajectory ($1296\times$★) or the final model weights of all fine-tuning trajectories ($1\times$★) as in (Ramé et al., 2022b). **Ensemble** results average instead the model outputs ($1296\times$▲ and $1\times$▲). Finally, **large dropout** results were obtained like the baseline results but using a 90% dropout rate on the penultimate layer ($1296\times$◆).

