# OpenReview forum: "fine-tuning with very large dropout"
_ICLR.cc/2025/Conference — Submitted to ICLR 2025_

### Official Review · Reviewer_HNY5 · 2024-11-01

**Soundness:** 2
**Presentation:** 2
**Contribution:** 1
**Rating:** 3
**Confidence:** 5

**Summary:**

The current manuscript challenges the traditional assumption that training and testing data share the same distribution, which is increasingly unrealistic in real-world applications. To handle diverse data distributions, recent work has used ensemble methods to create richer, more robust representations than those optimized for single-distribution performance. Instead, the authors propose fine-tuning large pre-trained models with large dropout rates, achieving better performance on OOD tasks. They compare the effective of their method with ensembles and weight-averaging methods.

**Strengths:**

This paper argues that the traditional assumption in ml—that training and testing data come from the same distribution—is no longer realistic. To handle real-world scenarios involving multiple, shifting data distributions, recent studies have highlighted the need for representations that are "richer" than those optimized for single-distribution performance.  and they suggest fine-tuning a large trained model via large dropout rates.

**Weaknesses:**

**w1: novelty**: my major concern is the limited novelty of the proposed approach, as it closely resembles existing methods without introducing substantial new concepts or techniques.

**w2:limited experiments and results:** even the experimental setup is minimal, with limited datasets and benchmarks, which weakens the generalizability and impact of the findings. Expanding the experiments to include a broader range of datasets and baselines would strengthen the evaluation.

**w3:Lack of theoretical discussion:** the paper lacks a theoretical foundation or in-depth discussion that could clarify the underlying mechanisms or justify the observed empirical results.

**Questions:**

Please see the above weaknesses section.

---

> ### Author Response · Authors · 2024-11-19
> **Response to reviewer HNY5**
>
> We highly recommend that Reviewer HNY5 read our [general response](https://openreview.net/forum?id=Fvfs0HPuKl&noteId=1srqKi5Pwc) to gain a better understanding of this work.
>
> ---
>
> ### Question 1: lack of novelty, because this paper doesn’t introduce substantial new concepts or techniques.
>
> To start with, substantial new concepts do not necessarily lead to novelty.
>
> Please allow me to ask two questions to justify the novelty of this work:
>
> - Before this paper, did the community know a simple very-large dropout on the penultimate layer outperformed ensembling, weight averaging, and other .xxx methods on ood?
>
> - Before this paper, would you even try to fine-tune with dropout rate >= 0.9 and explore the possibility of very-large dropout approach on ood generalization?
>
> Answering these two questions will justify the novelty of this work. To the best of our knowledge,  the answer to these two questions is likely “NO”
>
> ---
>
> ### Question 2: reviewer HNY5 feels the experimental setup is minimal.  And thus ask for more datasets and baselines.
>
> Please check our [general response 1 and 2](https://openreview.net/forum?id=Fvfs0HPuKl&noteId=1srqKi5Pwc).
>
> The experimental setup follows DomainBed, which is a widely used setup. Thus, it is not a “minimal” setup, but a “widely used” setup.
>
> DomainBed is not one dataset, but a set of datasets. Following [5], this paper uses 5 different datasets.
>
> This work compares very-large dropout with ensemble and weight averaging baselines.  The comparison between weight averaging and other baselines has been conducted in [9].
>
> As we stated in lines 202-208: "*Using these same datasets, Gulrajani & Lopez-Paz (2020) argue that plain Empirical Risk Minimization (ERM) equals and often betters the o.o.d. performance of purposefully designed methods, such as CORAL (Sun & Saenko, 2016), VREX (Krueger et al., 2021), and IRM (Arjovsky et al., 2019). More recently, Arpit et al. (2022), Cha et al. (2021), Ramé et al. (2022b), and Ramé et al. (2022a) find that ensemble and weight averaging methods consistently outperform the o.o.d. performance of ERM. We now show that fine-tuning with very large dropout outperforms the o.o.d. performance of these state-of-the-art methods.*"
>
> We can, of course, reference the numbers of these methods. But that only make the paper wordy.
>
> ---
>
> ### Question 3: Lack of theoretical discussion.
>
> Please refer to our [general response 1](https://openreview.net/forum?id=Fvfs0HPuKl&noteId=1srqKi5Pwc) for more information.
>
> ---
>
> [5] Rame, Alexandre, Matthieu Kirchmeyer, Thibaud Rahier, Alain Rakotomamonjy, Patrick Gallinari, and Matthieu Cord. "Diverse weight averaging for out-of-distribution generalization." Advances in Neural Information Processing Systems 35 (2022): 10821-10836.
>
> [9] Cha, Junbum, Sanghyuk Chun, Kyungjae Lee, Han-Cheol Cho, Seunghyun Park, Yunsung Lee, and Sungrae Park. "Swad: Domain generalization by seeking flat minima." Advances in Neural Information Processing Systems 34 (2021): 22405-22418.

---

> > ### Comment · Reviewer_HNY5 · 2024-11-24
> >
> > I have carefully considered the feedback from other reviewers, as well as the author's rebuttal, general responses, and their responses to other reviewers. While I appreciate the authors' thorough and thoughtful responses, I am not convinced by the explanations provided, and will not change my score! I would encourage the authors to explore a more comprehensive experimental design suggested by other reviewers and to reflect on theoretical insights from statistical learning theory. Incorporating these changes would improve the paper and make it more interesting for the ML community.

---

> ### Author Response · Authors · 2024-11-21
>
> Dear reviewer HNY5
>
> Thank you for taking the time to review our paper and provide valuable feedback.
>
> We have carefully considered your comments and have provided detailed responses to each of your points. We hope that our answers adequately address your concerns and provide a clearer understanding of our work.
>
> If you have any further questions or concerns, please do not hesitate to leave a comment.

---

> ### Author Response · Authors · 2024-11-24
>
> Thank you, Reviewer HNY5, for your thoughtful feedback. We greatly value the opportunity to enhance understanding between authors and reviewers, beyond just paper-review scores. To facilitate this understanding, we posed several questions in our initial response, seeking clarity on certain points. Unfortunately, we were unable to find clear answers in your [response](https://openreview.net/forum?id=Fvfs0HPuKl&noteId=NS6STaVVHv).
>
> We would like to reiterate our initial questions and sincerely hope to receive clear answers from you, which will enable us to improve our paper effectively.
>
> ---
> **Novelty Concerns**: You raised a question about the novelty of our work. We posed two questions in response:
>
> - *"Before this paper, did the community know that a simple very-large dropout on the penultimate layer outperformed ensembling, weight averaging, and other methods on OOD?*"
> - *"Before this paper, would you have considered fine-tuning with a dropout rate >= 0.9 and exploring the possibility of a very-large dropout approach on OOD generalization?"*
>
> To the best of our knowledge, the answer is "No." **Could you kindly provide an answer from your perspective?**
>
> ---
>
> **Experimental Setup**: You commented that our experimental setup is *"minimal"*. We clarified that the DomainBed benchmark is widely used, and our work compares two architectures (ResNet, Transformer), three pre-trained models, five OOD tasks, and thousands of fine-tuning configurations. The superior performance of our proposed very-large dropout approach is evident. Reference [9] has compared other baselines (e.g., CORAL, VREX, IRM) with our baseline (ensemble, weight-averaging), demonstrating the superior performance of our baseline. We can also reference the CORAL, VREX, IRM baselines in [9].
>
> We believe our experiment is sufficiently comprehensive to support our claims. We found Reviewer HNY5's suggestion to *"explore a more comprehensive experimental design"* in the response. We truly want to ensure the experimental design in this work is comprehensive.
>
>  **Could you kindly point out which parts of our current experimental design make it less comprehensive / minimal / fail-to-support-our-claim?**
>
> ---
>
> **Theoretical Discussion**: Reviewer HNY5 further pointed out that this work lacks theoretical discussion.
>
> In our response, we clearly stated that this is an experimental work. It's worth noting that ICLR accepts experimental works, which aligns with the nature of our submission.
>
> We appreciate your guidance and look forward to your insights, which will be invaluable in refining our work.

---

### Official Review · Reviewer_Lz8E · 2024-11-02

**Soundness:** 3
**Presentation:** 3
**Contribution:** 2
**Rating:** 5
**Confidence:** 4

**Summary:**

To improve the ensembling performance of weight averaging, the paper proposes to employ a large dropout rate at the penultimate layer to obtain rich representations with better OOD performance.

**Strengths:**

1. The training scheme of a large dropout is interesting.
2. The proposed large dropout obtains superior performance than previous methods.

**Weaknesses:**

1. The main problem of the paper does not provide a deep analysis of why large dropout is helpful. It simply provides some comparing experiments to demonstrate its effectiveness. The authors should provide a formal definition of the achieved "richer representation" (i.e., quantitative comparisons). Also, some visualizations also can be provided for a better analysis.
2. Since the experiments of the proposed method are conducted on the DomainBed, comparing them to the existing methods is also required.
3. The deep analysis of different behaviors of Weight Averaging, Deep Ensemble, and the proposed method on the ID and OOD performance is required for Figure 7.
4. Why not adopt the large dropout in the deeper layers?
5. The number of ensembling members is provided (1296? which seems too large for the practice). The influence of the number should also be considered.

**Questions:**

See Weaknesses.

---

> ### Author Response · Authors · 2024-11-19
> **Response to reviewer Lz8E**
>
> We appreciate Reviewer Lz8E's positive feedback on the superior performance of our proposed large dropout approach.
>
> ---
>
> ### Question 1: Could you add additional theoretical results or visual analysis?
>
> Please refer to our [general responses 1 and 2](https://openreview.net/forum?id=Fvfs0HPuKl&noteId=1srqKi5Pwc) for more information.
>
> ---
>
> ### Question 2: Compare the proposed method with other existing methods in the DomainBed benchmark.
>
> We have already done so. Please see lines 202-208, which state: "*Using these same datasets, Gulrajani & Lopez-Paz (2020) argue that plain Empirical Risk Minimization (ERM) equals and often betters the o.o.d. performance of purposefully designed methods, such as CORAL (Sun & Saenko, 2016), VREX (Krueger et al., 2021), and IRM (Arjovsky et al., 2019). More recently, Arpit et al. (2022), Cha et al. (2021), Ramé et al. (2022b), and Ramé et al. (2022a) find that ensemble and weight averaging methods consistently outperform the o.o.d. performance of ERM. We now show that fine-tuning with very large dropout outperforms the o.o.d. performance of these state-of-the-art methods.*"
>
> We can reference the numbers of these methods because the comparison between these methods and weight-averaging & ensemble has been conducted in related works [9].
>
> ---
>
> ### Question 3: Analyze the reason why the IID performance of the very-large dropout approach doesn’t always outperform weight averaging and ensemble (in appendix figure 7).
>
> Please refer to our [general response 3](https://openreview.net/forum?id=Fvfs0HPuKl&noteId=1srqKi5Pwc) for more information.
>
> ---
>
> ### Question 4: Why not adopt the large dropout in the deeper layers?
>
> Please refer to our [general response 4](https://openreview.net/forum?id=Fvfs0HPuKl&noteId=1srqKi5Pwc) for more information.
>
> ---
>
> ### Question 5: If the number of ensembling members is 1296, it is too large.
>
> The number of ensemble members is actually 2 (learning rate) x 3 (weight decay) = 6, not 1296. Therefore, it is not too large.
>
> ---
>
> [9] Cha, Junbum, Sanghyuk Chun, Kyungjae Lee, Han-Cheol Cho, Seunghyun Park, Yunsung Lee, and Sungrae Park. "Swad: Domain generalization by seeking flat minima." Advances in Neural Information Processing Systems 34 (2021): 22405-22418.

---

> ### Author Response · Authors · 2024-11-21
>
> Dear reviewer Lz8E
>
> Thank you for taking the time to review our paper and provide valuable feedback.
>
> We have carefully considered your comments and have provided detailed responses to each of your points. We hope that our answers adequately address your concerns and provide a clearer understanding of our work.
>
> If you have any further questions or concerns, please do not hesitate to leave a comment.

---

> > ### Comment · Reviewer_Lz8E · 2024-11-25
> >
> > Thanks for the authors' effort in the rebuttal. However, the theoretical analysis of the method is still not provided. The author actually should explain with a formal theory why the dropout can facilitate the "rich feature" learning process instead of simply citing the related work. Therefore, I will keep my rate.

---

> ### Author Response · Authors · 2024-11-25
>
> Thanks for reviewer Lz8E's quick reply.
>
> After our response, reviewer Lz8E's primary concern is that this **empirical work** doesn't contain a **"formal theory"**. We are happy to see that our responses addressed reviewer Lz8E's other [question 2-5](https://openreview.net/forum?id=Fvfs0HPuKl&noteId=ylRzS8h1nS).
>
> Please note that this is an empirical work on the early-stage of a counter-intuitive problem. Because it is the early-stage of a "unfamilar" problem, establishing a consistent & helpful **"formal theory"** requires a lot of empirical "observations" as support.   This work serves as one of these essential empirical observations.

---

### Official Review · Reviewer_RZvb · 2024-11-03

**Soundness:** 2
**Presentation:** 2
**Contribution:** 2
**Rating:** 6
**Confidence:** 4

**Summary:**

In this work the authors make one simple observation: when fine-tuning in o.o.d. setups, using a very high dropout can yield surprisingly good performance. The evaluation is performed on popular downstream datasets and common architectures.

**Strengths:**

- The quantitative evaluation (for obtaining the main results) is quite comprehensive.
- The message is delivered clearly.
- The empirical result is quite surprising.

**Weaknesses:**

- There is a gap from the expected theory. In the limit of large sampling, there should not be any difference in terms of performance when varying the dropout probability (a-priori, large dropout should make convergence harder, when untouching the learning rate). The empirical observation in the specific setup chosen is interesting, but right now, considering it is the only contribution, it is felt there is a lack of grounding to allow practitioners to employ this solution without running an extensive grid search.
- It seems that there is a lot of engineered choices related to learning rate scaling at layer's scale, dropout etc.
- Some statements are overly (and maybe, unnecessarily) strong. For example, "It is impossible today to pretend that the practice of machine learning is compatible with the idea that training and testing data follow the same distribution": there are many cases in which inference happens in distribution.

**Questions:**

- How is the proposed idea performing on more efficient architectures, like MobileNet?
- Is the same approach valid for non CV-based tasks, like NLP? It is felt the high dropout is successful where architectures are notoriously over-parametrized (as it is for CV).
- How are the learning hyper-parameters found? Has a validation set been used?

---

> ### Author Response · Authors · 2024-11-19
> **Reviewer RZvb**
>
> We encourage Reviewer RZvb to read the [general response](https://openreview.net/forum?id=Fvfs0HPuKl&noteId=1srqKi5Pwc) to gain a comprehensive understanding of this work.
>
> ---
>
> ### Question 1: In the limit of large sampling, this work conflicts with the expected theory. Thus practitioners need an extensive grid search to employ this solution.
>
> The assumption of "large sampling" is not applicable to our work. This work focuses on "limited samples" in the OOD fine-tuning area. Consequently, theories based on large sampling assumptions do not apply here.
>
> Regarding the claim of an extensive grid search, it is unfounded. Our very-large dropout approach involves only one hyper-parameter: the dropout rate. Figure 2 demonstrates that the relationship between dropout rate and OOD accuracy is simple and smooth, indicating that our approach is not sensitive to the choice of dropout rate. Therefore, there is no extensive grid search involved.
>
> ---
>
> ### Question 2: Reviewer RZvb points out that “there is a lot of engineered choices related to learning rate scaling at layer's scale, dropout etc.”
>
> This is incorrect. “Learning rate scaling at layer’s scale”  is merely a method for comparison, not a hyper-parameter of our proposed very-large dropout approach.
>
> In fact, our work aims to minimize engineering choices, such as learning rate and weight decay. The bar plot in Figure 1 illustrates various hyper-parameter configurations, comparing the worst hyper-parameters of very-large dropout with the best hyper-parameters of other methods.
> Therefore, it is incorrect to write that "a lot of engineered choices” are involved.
>
> ---
>
> ### Question 3: Some statements are overly (and maybe, unnecessarily) strong. For example, "It is impossible today to pretend that the practice of machine learning is compatible with the idea that training and testing data follow the same distribution": there are many cases in which inference happens in distribution.
>
> The claim “*there are many cases in which inference happens in distribution.*” does not conflict with  the statement  "*It is impossible today to pretend that the practice of machine learning is compatible with the idea that training and testing data follow the same distribution*".
>
> There are carefully designed cases of in-distribution. In the meanwhile, there are more cases of out-of-distribution in the wild (e.g. each in-distribution case can be easily transformed into multiple ood cases by changing spurious features).
>
> Reviewer RZvb is talking about the existence of in-distribution. Our sentence states that not all cases and not even a majority of cases are in-distribution.
>
> ---
>
> ### Question 4: How does it work on other tasks and other architectures?
>
> Please refer to our [general responses 1 and 2](https://openreview.net/forum?id=Fvfs0HPuKl&noteId=1srqKi5Pwc).
>
> ---
>
> ### Question 5: How are the learning hyper-parameters found? Has a validation set been used?
> This work utilizes the DomainBed suite [7] and weight-averaging as a baseline [8], adhering to the setups in these references. As stated in lines 282-283, "Following Gulrajani & Lopez-Paz (2020), we prevent overfitting by early-stopping at the best i.i.d. performance." We use an IID validation set as per the setups in [7].
>
> ---
>
> [7] Gulrajani, Ishaan, and David Lopez-Paz. "In search of lost domain generalization." arXiv preprint arXiv:2007.01434 (2020).
>
> [8] Rame, Alexandre, Matthieu Kirchmeyer, Thibaud Rahier, Alain Rakotomamonjy, Patrick Gallinari, and Matthieu Cord. "Diverse weight averaging for out-of-distribution generalization." Advances in Neural Information Processing Systems 35 (2022): 10821-10836.

---

> > ### Author Response · Authors · 2024-11-25
> >
> > Thanks reviewer RZvb for your feedback. We are pleased to see that our previous response addressed some of your concerns, and we appreciate the time you took to review our work. We are also grateful for your positive comments on our submission. Your encouragement is much appreciated.

---

> ### Author Response · Authors · 2024-11-21
>
> Dear reviewer RZvb
>
> Thank you for taking the time to review our paper and provide valuable feedback.
>
> We have carefully considered your comments and have provided detailed responses to each of your points. We hope that our answers adequately address your concerns and provide a clearer understanding of our work.
>
> If you have any further questions or concerns, please do not hesitate to leave a comment.

---

> > ### Comment · Reviewer_RZvb · 2024-11-25
> >
> > After rebuttal, I feel my concerns are partially addressed. I really appreciated the attention devoted by the authors in addressing the reviewer's comments. I still disagree with some of the author's comments, but overall I see much more clearly the paper's point, and I invite the authors to keep updating the paper in this spirit. I am happy to raise my score to a marginal accept.

---

### Official Review · Reviewer_hP51 · 2024-11-03

**Soundness:** 3
**Presentation:** 3
**Contribution:** 2
**Rating:** 5
**Confidence:** 3

**Summary:**

The submission proposes a technique for fine-tuning pre-trained networks: apply a very large dropout rate (90%) on the penultimate layer of residual networks. This approach is shown to be effective at making fine-tuned networks more robust to an OOD dataset, when compared to two existing approaches.

**Strengths:**

Originality: Very high dropout rates have not been explored because they tend not to work well when training from scratch. Exploring it in the context of modern residual networks and fine-tuning with an emphasis on evaluating OOD robustness is original, to my knowledge.

Clarity: The paper is very clearly written, and is easy to follow.

Quality: The ideas are more or less technically sound, building on previous observations. The experiments are adequate to support the claims made — high dropout is more effective for OOD generalization for a fine-tuned model relative to related ensemble methods.

Significance: The results appear to be consistent, though marginal most often, improvements over the selected baselines. Given that the proposed method is simple and lightweight makes it practically interesting.

**Weaknesses:**

While the intuitions motivating the method are interesting and seem to play the role of establishing weight-averaging and ensemble as the natural baselines, it is not very clear that these intuitions are sufficient to narrow the scope of comparison to only the selected baselines.

It seems that performance is somewhat sensitive to the precise value of the dropout rate — 90%. It is not clear that this rate was chosen without looking at test-time performance.

It is not fully clear to me why applying large dropout only to the penultimate layer is necessarily the best implementation, given the motivation. Does performance suffer if such dropout is applied everywhere, and not in a “dimension-aligned” fashion?


L296-298 says “the i.i.d. performance of the large dropout method lags behind that of ensembles, revealing that the o.o.d. performance improvement is not a secondary effect of some i.i.d. performance improvement (i.e. the o.o.d. performance gaps in o.o.d. Figure 1 do not come from i.i.d. overfitting/underfitting.).” In general, one wishes to retain good IID performance while also improving upon OOD numbers. On balance, might a practitioner prefer one of the baselines?

Minor:

L150-L153 says: `We focus on residual networks because fine-tuning has been found to hardly change the inner layers of non-residual networks (Raghu et al., 2019, fig 2). In contrast, skip connections in residual networks expose the inner block features in such a manner that the fine-tuning process can utilize these features in a near-linear way (Evci et al., 2022).` The first citation seems to be about MAML, which is a specific setting and likely does not extend to all fine-tuning settings. The point in that work seems to be more about what meta-learning does in terms of feature reuse and does not make generic observations about fine-tuning in non-residual networks. The second citation doesn’t seem to make any claims about the usefulness of intermediate features being specific to models using skip connections. ResNets and ViTs are top classifiers currently, so it might not be necessary to motivate the choice of models here anyway.

L319-L321 says `When the fine-tuning dataset size approaches the pre-training dataset size, the difference between fine-tuning and training from scratch becomes less clear, the linear connectivity property disappears, the linear approximation perspective on fine-tuning no longer holds, and the optimal dropout rate falls sharply.` Has it been demonstrated in prior work that the linear connectivity property disappears when the dataset sizes get close (i.e. is there a citation for this claim)?

L447 says `Despite all this technology, advanced fine-tuning of a pretrained RESNET #1 (2nd column) barely matches the performance of naive fine-tuning on RESNET #2 (3rd column).` Just a nit, but usually one does not say “A barely matches B” when A > B.

Typos:

L129: “setup is a commonly used” —> drop “a”

**Questions:**

Please see above.

---

> ### Author Response · Authors · 2024-11-19
> **Response to Reviewer hP51**
>
> We highly recommend that reviewer hP51 read our [general response](https://openreview.net/forum?id=Fvfs0HPuKl&noteId=1srqKi5Pwc), as most of their concerns are addressed there.
>
> ---
> ### Question 1: The choice of baseline methods in this paper is narrow.
> Please refer to our [general responses 1 and 2](https://openreview.net/forum?id=Fvfs0HPuKl&noteId=1srqKi5Pwc) for a detailed explanation.
>
> ---
> ### Question 2: Reviewer hP51 concerns the proposed very-large dropout method is sensitive to dropout rate hyper-parameter = 0.9.
> No, this paper provides the Ablation of dropout rate in Figure 2. First, the curve of dropout rate vs OOD acc is simple & smooth, which makes it easy to search hyper-parameters. Second, both dropout rates 0.9 and 0.95 provide good OOD performance.
>
> ---
> ### Question 3: Why does the proposed method help OOD generalization? Why apply dropout on the penultimate layer, instead of everywhere?
> Please read our [general responses 1 and 4](https://openreview.net/forum?id=Fvfs0HPuKl&noteId=1srqKi5Pwc) for an explanation.
>
> ---
> ### Question 4: Why does IID performance drop in certain datasets?
> Please read our [general response 3](https://openreview.net/forum?id=Fvfs0HPuKl&noteId=1srqKi5Pwc) for an explanation.
>
> ---
> ### Question 5: Reviewer hP51 points out that Raghu et al., 2019 is a case about MAML and likely does not extend to all fine-tuning settings.
>
> Thanks for your suggestion. There are also other works [4] that demonstrate a close phenomenon as Raghu et al., 2019 figure 2, but in a more general setting. We will cite [4] in the final version.
>
>
> ---
> ### Question 6: Reviewer hP51 believes that Evci et al., 2022 “doesn’t seem to make any claims about the usefulness of intermediate features being specific to models using skip connections”. Thus ###
>
> Question whether this work should use Evci et al., 2022 to motivate the choice of models.
> The claim that “skip connections in residual networks expose the inner block features” is supported by Equation 1 in our paper. Evci et al., 2022 experimentally demonstrate the usefulness of inner block features on networks with skip connections (resnet, transformer), and thus support the claim above in the experimental view. We use Evci et al., 2022 to motivate the choice of ResNet and Transformer because they use these two choices and support our claim.
>
> ---
> ### Question 7: Has it been demonstrated in prior work that the linear connectivity property disappears when the dataset sizes get close (i.e., is there a citation for this claim)?
>
> The size of the pretraining dataset (in this work) is large compared with the fine-tuning dataset size. As the fine-tuning dataset gets larger, one needs more fine-tuning iterations (line 183) or other aggressive hyper-parameters (such as large learning rate) to pursue better performance. Many weight-averaging works, e.g., [5] (section E.1.4) and [6], show that aggressive fine-tuning hyper-parameters breaks the linear connectivity property. We will add citations for this claim in the final version.
>
>
> ---
> ### Question 8: “A barely matches B” -> “A matches B”. “setup is a commonly used” —> drop “a”.
>
> Thanks for pointing them out. We will fix them in the final version.
>
>
> ---
>
> [4] Zhang, Chiyuan, Samy Bengio, and Yoram Singer. "Are all layers created equal?." Journal of Machine Learning Research 23, no. 67 (2022): 1-28.
>
> [5] Rame, Alexandre, Matthieu Kirchmeyer, Thibaud Rahier, Alain Rakotomamonjy, Patrick Gallinari, and Matthieu Cord. "Diverse weight averaging for out-of-distribution generalization." Advances in Neural Information Processing Systems 35 (2022): 10821-10836.
>
> [6] Wortsman, Mitchell, Gabriel Ilharco, Samir Ya Gadre, Rebecca Roelofs, Raphael Gontijo-Lopes, Ari S. Morcos, Hongseok Namkoong et al. "Model soups: averaging weights of multiple fine-tuned models improves accuracy without increasing inference time." In International conference on machine learning, pp. 23965-23998. PMLR, 2022.

---

> ### Author Response · Authors · 2024-11-21
>
> Dear reviewer hP51
>
> Thank you for taking the time to review our paper and provide valuable feedback.
>
> We have carefully considered your comments and have provided detailed responses to each of your points. We hope that our answers adequately address your concerns and provide a clearer understanding of our work.
>
> If you have any further questions or concerns, please do not hesitate to leave a comment.

---

> ### Comment · Reviewer_hP51 · 2024-11-24
> **Post-rebuttal follow-up**
>
> I thank the authors for their thorough response. Having read it, and considered the points being raised by the other reviewers, I am inclined to keep my rating. In particular,
> * while it is true that purely empirical results are within the scope of ICLR, a paper that presents itself as purely empirical without very clear reasoning would need to prove itself much more thoroughly. DomainBed is arguably at this point a somewhat out-dated benchmark, with more relevant ones such as NICO++ or WILDS available, where one can more convincingly illustrate the empirical merit of an idea. One must also consider the current era we live in, where massively trained models can likely solve most of the DomainBed flavour of tasks.
> * `No, this paper provides the Ablation of dropout rate in Figure 2. First, the curve of dropout rate vs OOD acc is simple & smooth, which makes it easy to search hyper-parameters. Second, both dropout rates 0.9 and 0.95 provide good OOD performance.` There are only 4 points of evaluation in Fig 2. While the performances at 0.9 and 0.95 are fairly close for 3 of the 4 datasets, and more significant for the challenging TerraIncognita dataset, the overall improvements over the baselines are not so high that one can ignore the differences.
> * The analogy of OOD generalization to the IID train/test setup is not convincing, and is in fact a limitation with the narrative in this area of work. One cannot know how OOD a deployment is going to be, and that the model will never be evaluated in an IID setting. In fact, it feels a much more likely scenario that models are generally going to be evaluated in IID settings, but may face OOD settings at varying rates depending on circumstances.

---

> ### Author Response · Authors · 2024-11-24
>
> Thank you, Reviewer hP51, for your thoughtful feedback. We appreciate the opportunity to address your concerns and provide further clarification.
>
> ---
>
> Regarding the **benchmark discussion**, DomainBed [1], WILD [2], and NICO++ [3] were all released between 2020 and 2022. They share a similar dataset structure (multiple training environments), and benchmark a comparable set of methods (e.g., IRM, ERM, VREX, GroupDRO, etc). Therefore, it is challenging to label one as out-dated while considering the others current.
>
> More importantly, these benchmarks are designed to evaluate whether a machine learning algorithm can learn interesting properties, rather than focusing solely on absolute performance, such as that achieved by "massively trained models." Training directly on test data would yield the best performance, but it would not be interesting.
>
>
> Thus, the statement, *"One must also consider the current era we live in, where massively trained models can likely solve most of the DomainBed flavor of tasks,"* does not imply that evaluating a machine learning algorithm on DomainBed is less convincing. DomainBed benchmark evaluates the ood generalization problems that are still challenging to solve.
>
> ---
>
> In response to Reviewer hP51's inquiry about **the sensitivity of dropout parameters**, we present two arguments:
>
> - *"As shown in Figure 2, the curve of dropout rate versus OOD accuracy is simple and smooth, facilitating easy dropout hyper-parameter search.*"
> - *"Both dropout rates of 0.9 and 0.95 yield good OOD performance.*"
>
> Reviewer hP51's observation that *"the performances at 0.9 and 0.95 are fairly close for 3 of the 4 datasets"* supports our claim that the very-large dropout approach in this work is not sensitive to dropout hyper-parameters. More importantly, the former argument shows that dropout rate is not a sensitive hyperparameter in this work.
>
> ---
>
> Reviewer hP51 further notes that *"and more significant for the challenging TerraIncognita dataset."* We argue that it is not inherently "challenging"; rather, TerraIncognita contains more training examples. Please refer to, lines 318-338 and the DomainNet dataset experiment in Figure 3 for further clarification.
>
> ---
>
> Regarding **the analogy of OOD generalization to the IID train/test setup**, Reviewer hP51 mentions that it is *"not convincing, and is in fact a limitation with the narrative in this area of work."* We would like to clarify that this analogy is general and not exclusive to our work.
>
> To elucidate, consider a simple question in IID: *"If we model the training data with a Dirac delta function, does the model provide any insight into IID test data?  answer: **No**."*
>
> Similarly, in OOD: *"If we model training data from distribution A, does the model offer any insight into test data from arbitrarily distribution B?  answer:  **No** "*
>
> We can have a "**Yes**" answer when we eliminate "**arbitrarily**" in OOD (building connections between distirbution A and B) and "**Dirac delta function**" in IID (building connections between IID train data and IID test data).
>
> [1] Gulrajani, Ishaan, and David Lopez-Paz. "In search of lost domain generalization." arXiv preprint arXiv:2007.01434 (2020).
>
> [2] Koh, Pang Wei, et al. "Wilds: A benchmark of in-the-wild distribution shifts." International conference on machine learning. PMLR, 2021.
>
> [3] https://arxiv.org/abs/2204.08040

---

### Official Review · Reviewer_azwu · 2024-11-04

**Soundness:** 2
**Presentation:** 3
**Contribution:** 2
**Rating:** 6
**Confidence:** 4

**Summary:**

In the paper, the authors propose a novel, effective fine-tuning method to fix reduced performance on out-of-distribution (OOD) data. In essence, they begin with a model pretrained on a larger dataset and then fine-tune it on a significantly smaller set of labelled data. Given the well-known tendency of modern neural networks to perform poorly on OOD samples, the proposed method aims to reduce this performance gap. The approach is straightforward: it fine-tunes the pretrained model with applying dropout (with high dropout rate) to the penultimate layer. This method improves the final model's performance on OOD data across multiple benchmarks (PACS, VLCS, Office Home, Terra Incognita), surpassing popular baselines like model ensembling and weight averaging in classification accuracy. Furthermore, it introduces significantly less computational and memory overhead and enables faster training than conventional ensembles. The proposed approach is also easy to apply across different architectures.

**Strengths:**

* The paper is clearly written and easy to follow, with an intuitive and easily understandable idea. The related work section provides a sufficient discussion of existing approaches for generating rich feature representations suitable for OOD data.

* The method is straightforward to apply, requiring only a pretrained model and a simple fine-tuning procedure that is easy to implement and computationally efficient. This approach does not change the inference of the final model (compared to vanilla model baseline), making it significantly more computationally efficient than ensembling-based alternatives.

* The method achieves strong results across various classification benchmarks.

**Weaknesses:**

* The experiments are limited to classification tasks on fairly standard datasets, leaving it unclear whether this approach with high dropout rates generalizes effectively to other tasks, such as segmentation or even text classification, or to other architectures, like detection models and etc. This raises questions about the scalability and broader applicability of the proposed method beyond image classification. Can other modalities/tasks be added?

* While the paper offers a high-level explanation of why applying high dropout during fine-tuning could improves generalization (by generating more "weakly relevant" features), it is not entirely clear why this should occur from a formal standpoint. Consequently, the high dropout rate forces the model to distribute information more evenly across dropped features. However, why a more even distribution would result in more "weakly relevant" features remains unclear. A more detailed discussion would be beneficial.

* As shown in Figure 7 of the appendix, the ensembling approach consistently improves performance on in-distribution data, whereas the proposed high-dropout approach does not always yield similar benefits (and sometimes even reduces performance compared to the baseline, which is somewhat expected). It appears that overly aggressive dropout can significantly undermine performance due to excessive randomness. Could more structured methods, such as Masksembles [1], address this issue? Such methods might also overcome the problem of training a model from scratch with high dropout (which often fails to converge due to randomness) by introducing a more structured way to drop features.

[1] Durasov, Nikita, et al. "Masksembles for uncertainty estimation." CVPR 2021.

**Questions:**

* Can the proposed high-dropout method generalize to tasks and architectures beyond classification, such as segmentation, text classification, or detection? Can you provide such evaluations?

* Why should evenly distributed information across dropped features lead to “weakly relevant” features? Is further theoretical support available?

* The underperformance of ensembles is somewhat surprising. A potentially useful baseline could involve training ensembles with a shared classification head, where all layers except the final one remain distinct for each model in the ensemble. This setup would encourage the ensemble to generate features from the same domain, allowing for feature averaging, which could improve OOD performance. Would that resolve the underperformance issue?

* Could structured dropout methods, such as Masksembles, help address the issue of aggressive dropout leading to worse performance on in-distribution data? Could it help train models from scratch with high dropout rates without compromising generalization?

---

> ### Author Response · Authors · 2024-11-19
> **Response to Reviewer azwu**
>
> We encourage reviewer azwu to read our [general response](https://openreview.net/forum?id=Fvfs0HPuKl&noteId=1srqKi5Pwc) first.
>
> ---
> ### Question 1: Why does the proposed approach work? And thus suggest evaluating the proposed approach on other tasks and architectures?
>
> Please refer to our [general responses 1 and 2](https://openreview.net/forum?id=Fvfs0HPuKl&noteId=1srqKi5Pwc) for a detailed explanation.
>
> ---
> ### Question 2: Concerns about the dropping IID performance in certain datasets.
>
> Please see our general [response 3](https://openreview.net/forum?id=Fvfs0HPuKl&noteId=1srqKi5Pwc) for an explanation.
>
> ---
>
> ### Question 3: Suggest using "structured dropout methods" to avoid the dropping IID performance in certain datasets.
>
> We appreciate reviewer azwu's suggestion. Please check our [general response 3](https://openreview.net/forum?id=Fvfs0HPuKl&noteId=1srqKi5Pwc). In short, IId performance in the OOD generalization area is analogous to training dataset performance in traditional IID machine learning. Thus IID performance is not the primary concern in the OOD generalization area.
>
> ---
> ### Question 4: Suggest increasing the ensemble baseline (OOD) performance by training a model with one joint classifier and multiple separate backbones.
>
> We thank reviewer azwu for the suggestion. However, this approach has been shown to lead to poor OOD performance. As demonstrated in related work [2], training a model with one joint classifier and multiple separate backbones can harm rich representation and result in suboptimal OOD performance.
>
> ---
> ### Question 5: Why should evenly distributed information across dropped features lead to "weakly relevant" features? Is further theoretical support available?
>
> To satisfy the learning objectives, a model needs to identify relevant features. Under the sparsity bias (either from SGD optimizer or L1/L2 regularization), a few "strongly relevant" features can be discovered by minimizing the learning objectives. With the existence of “strong relevant” features, other features ( “weakly relevant”) are not incrementally helpful in the training distribution.
>
> “*However, features that are not strongly relevant might nevertheless
> (a) be incrementally useful when the data follows a different distributions of interest, or
> (b) be useful under the training distribution when added to certain subsets of the other existing
> features instead of all of them (“weakly relevant”).*”(line 049-052)
>
>
> A very-large dropout masks a large fraction of features, making it possible for the model to learn from these weakly relevant features.
>
>
> [2] Zhang, J., Lopez-Paz, D., & Bottou, L. (2022, June). Rich feature construction for the optimization-generalization dilemma. In International Conference on Machine Learning (pp. 26397-26411). PMLR.

---

> > ### Author Response · Authors · 2024-11-26
> >
> > Thanks reviewer azwu for the constructive feedback and suggestion for acceptance. We appreciate your input and would like to address one important comment regarding **the IID-vs-OOD performance question**.
> >
> >
> > The goal of OOD generalization is to help board OOD distributions. *IID distribution is one distribution, while OOD distributions are substential amount of other distributions.* (many vs one). Ideally, we aim to perform well on both IID and OOD distributions, as suggested by reviewer azwu. However, if this ideal case is not achievable, our work prioritizes OOD distributions due to their larger number.
> >
> > We considered exploring possibilities, such as reviewer azwu's "structured dropout" suggestion, to achieve superior performance on both IID and OOD. Unfortunately, we found it challenging to do so without making assumptions about the features in the OOD tasks.
> >
> > The reason lies in the difference between the training dataset (IID) and test dataset (OOD), which use different groups of features to predict the target label. For instance, the training data might use (color, shape) features, while the test data uses (shape, texture) features.
> >
> > The optimal IID performance is achieved by **jointly** utilizing all relevant features, i.e. "color" and "shape". However, this approach fails on test data, which uses a different group of features (shape, texture). To perform well on the test data without knowing its specific features, we use all IID features **separately**, rather than **jointly**. This "separate" solution, of course, decreases IID performance.
> >
> > On the other hand, one can gain some hints about test data features by, for example, having multiple training distributions. However, in this work, we study the scenario with only one traing distribution.
> >
> > We hope this clarifies our position on the IID-vs-OOD performance question.
> >
> > Thank you again for your feedback and suggestions.

---

> ### Author Response · Authors · 2024-11-21
>
> Dear reviewer azwu
>
> Thank you for taking the time to review our paper and provide valuable feedback.
>
> We have carefully considered your comments and have provided detailed responses to each of your points. We hope that our answers adequately address your concerns and provide a clearer understanding of our work.
>
> If you have any further questions or concerns, please do not hesitate to leave a comment.

---

> > ### Comment · Reviewer_azwu · 2024-11-25
> > **Rebuttal Reply**
> >
> > I would like to thank the authors for their effort in writing a rebuttal and addressing my questions and concerns. *In short*, I believe this paper is a borderline submission, though I **lean toward assessing it positively**. The discussion below is intended primarily for the authors, providing some thoughts on how they could further develop the paper in future submissions or through extensions of the proposed approach. The paper's central idea—using a large dropout rate to fine-tune the model to improve generalization—is quite novel, and I have not seen this setup successfully applied in the literature. At the same time, I feel that many of the questions I raised in my original review were not fully addressed, although the authors made an effort to provide some answers, for which I am grateful.
> >
> > Below, I will respond to the authors' replies to my questions with my comments.
> >
> > ### Question 1: Why does the proposed approach work? Should it be evaluated on other tasks and architectures?
> >
> > I agree that the request by some reviewers for a theoretical justification of the proposed method is somewhat excessive. The method has an intuitive basis: the model learns features that are robust and equally important. A somewhat similar idea is used in regularization methods applied to model weights, where promoting weight equality often leads to better generalization and robustness. Therefore, while a theoretical justification would be appreciated, it is not strictly necessary. The method is intuitive and has demonstrated effectiveness on practical benchmarks.
> >
> > However, I strongly disagree with the authors' response regarding additional benchmarks and models. As I understand, the method currently addresses only image classification tasks, which is quite limited. A major issue with many recently proposed generalization methods is their inability to scale to other tasks (e.g., regression or pixel-wise tasks like segmentation). Methods that are confined to image classification are theoretically interesting but less impactful from a practical perspective. Thus, I do not agree that the evaluation in the paper is sufficient. Even the authors acknowledge that extending the evaluation would be beneficial.
> >
> > ### Questions 2–3: Concerns about dropping IID performance on certain datasets.
> >
> > Here, I also strongly disagree with the authors. The primary goal of domain generalization approaches is *not* solely to improve performance on OOD data but to enhance the model's performance *overall*, across all types of data. For example, when deploying a model in a production environment, we aim for strong performance on both ID and OOD data—not just OOD. The setup where a model is fine-tuned on a small in-distribution set and then applied to OOD data while ignoring ID performance is plausible but not too practical. Therefore, I strongly advise against claiming in the paper that "IID performance is not the primary concern in the OOD generalization area."
> >
> > This brings me to the point about more structured dropout approaches. The reason I mentioned such methods is that standard dropout does not appear to be an optimal choice, as it often provides weaker predictions compared to structured versions or even ensembling in other tasks (for example, uncertianty estimation). While the authors seem to disagree with this point, there are multiple works that observe this trend. Other *better* methods than MC-Dropout could potentially yield superior performance on OOD data, which is why I originally suggested exploring them in my review. The method variation with MC-Dropout is still valuable, it just can be improved with more sophisticated approaches.
> >
> > For the remaining questions, I find that the authors provide reasonable explanations and answers.
> >
> > While I consider this work to be a positive borderline submission—offering an interesting and novel idea but with room for improvement—I am inclined to increase my score to 6. In general, I would be pleased to see this submission accepted, and I encourage the authors to continue refining and extending their approach. I hope my suggestions will contribute to the further development of the paper.
> >
> > P.S.: The rebuttal would have been significantly more compelling if the authors had included numerical evaluations suggested by reviewers (e.g., replacing MC-Dropout layers with structured dropout approaches). This modification requires only a few lines of code and no changes to training or evaluation, just a replacement of layers. The absence of similar evaluations for other reviews may be a reason why some reviewers remain hesitant to upgrade their scores.
> >
> > Hope this helps!
> >
> > Best regards,
> > azwu

---

### Author Response · Authors · 2024-11-19
**General Response**

## Question 1: Why does the very-large-dropout approach help OOD generalization? Is there a theoretical result?

Our paper explains the reason behind the success of the very-large-dropout approach in OOD fine-tuning as "rich representation" (Lines 043-063). This concept is also discussed in related works [1,2,3]. We want to emphasize that our approach and the idea of rich representation are counter-intuitive compared to traditional in-distribution views. As such, it may be challenging for those familiar with in-distribution machine learning to accept this new perspective.

However, we believe that our experimental results objectively demonstrate the superior performance of the very-large-dropout approach and the existence of a new view that differs from traditional in-distribution views. Even if our explanations fail to convince you, the superior performance of very-large-dropout is objective and contributive. It's worth noting that ICLR accepts experimental works, which aligns with the nature of our submission.


---
## Question 2: How does it work on other tasks and other architectures?

This concern is closely related to the previous question, as our proposed approach is counter-intuitive and may be challenging to accept subjectively. Consequently, one may require more support or evidence to fully accept it.

We understand this concern and acknowledge that it is reasonable. However, we would like to highlight that our paper has already provided a comprehensive evaluation of the very-large-dropout approach on two widely used architectures (ResNet, Transformer), three pre-trained models, five OOD tasks, and thousands of fine-tuning configurations. We believe that this extensive evaluation provides sufficient evidence to support the claims made in our paper.

While we agree that more experiments would be beneficial, we must also consider the page limit of the paper. As a nine-page submission, we have made a concerted effort to provide a clear and concise presentation of our research, while also providing sufficient evidence to support our claims.

---

## Question 3: IID performance drops down! Is this a drawback of the proposed very-large dropout approach?

We understand the concern about the drop in IID performance. However, we would like to emphasize that our work focuses on out-of-distribution (OOD) generalization, as stated in the DomainBed Benchmark setup and the abstract.

To put this into perspective, let's consider an analogy with traditional IID machine learning:
- In the traditional IID setup, one does not really care about the training accuracy but cares about the test accuracy. Because the true test examples are not necessarily the same as the training examples.

- In this OOD setup, one does not really care about the IID test accuracy, but cares about the OOD accuracy. Because the true test distribution is not necessarily the same as the training distribution.

In this context, the drop in IID performance is not necessarily a drawback, but rather a consequence of prioritizing OOD generalization. Our experimental results demonstrate the effectiveness of the very-large-dropout approach in achieving superior OOD performance, which is the primary focus of our work.

---

## Question 4: Why apply dropout on the penultimate layer, instead of everywhere (all layers)?

Applying high dropout rates to intermediate layers can significantly alter the input to subsequent layers, making fine-tuning more challenging. As we mentioned in line 066-072, fine-tuning with very-large dropout relies on the "simplicity" (nearly convex) of the fine-tuning process. By applying dropout to the penultimate layer, we preserve the simplicity of the fine-tuning process, which is essential for our approach to be effective.

[1] Zhang, J., & Bottou, L. (2023, July). Learning useful representations for shifting tasks and distributions. In International Conference on Machine Learning (pp. 40830-40850). PMLR.

[2] Zhang, J., Lopez-Paz, D., & Bottou, L. (2022, June). Rich feature construction for the optimization-generalization dilemma. In International Conference on Machine Learning (pp. 26397-26411). PMLR.

[3] Chen, Y., Huang, W., Zhou, K., Bian, Y., Han, B., & Cheng, J. (2024). Understanding and improving feature learning for out-of-distribution generalization. Advances in Neural Information Processing Systems, 36.

---

### Meta-Review · Area_Chair_uLq5 · 2024-12-20

**Metareview:**

This paper shows that fine-tuning large pre-trained models with a very high dropout rate applied to the penultimate layer improves out-of-distribution (OOD) generalization. The authors explain it as providing "richer" feature representations beneficial for OOD tasks. The approach is computationally efficient, requires fewer resources than ensembles, and performs robustly across different architectures (e.g., ResNets, Transformers). The paper is well-written, making the method and results accessible.

While the observation in this paper is simple, interesting, and potentially of broad interest to the community, the paper falls short in both practicality and theoretical insights.

- Practicality: multiple reviewers raised the concern that the experiments are restricted to image classification, leaving open questions about the method’s applicability to other tasks (e.g., segmentation, NLP). The authors acknowledged this limitation but did not address it experimentally. In addition, multiple reviewers raised concerns about the method's drop in IID performance, which may limit its practicality in scenarios requiring both IID and OOD generalization.

- The paper lacks a rigorous theoretical explanation of the observed phenomenon. For example, the authors resort to the notion of "richer representations" as an explanation but this does not have a quantitative definition.

While not all of these issues are necessarily required for acceptance, addressing at least one of them in a significant way would greatly strengthen the paper. I encourage the authors to refine their manuscript along these lines for future submissions.

**Additional Comments On Reviewer Discussion:**

Concern: Lack of theoretical analysis of "rich representations."

Response: The authors argued that this work represents an early empirical observation of a counter-intuitive phenomenon and cited related works to provide context. However, the concern remains unresolved.

Concern: Limited applicability to tasks beyond image classification.

Response: The authors acknowledged this limitation but did not extend their experiments beyond the current scope.

Concern: Trade-offs in IID performance.

Response: The authors justified prioritizing OOD generalization over IID performance, likening it to prioritizing test performance in traditional IID setups. This explanation was not entirely convincing to some reviewers.

While the rebuttal addressed many points, most reviewers remained critical of the limited theoretical grounding and lack of adequate experiments. While reviewers have slightly different opinions on whether the paper could be accepted due to these shortcomings, the overall assessment leaned toward rejection.

---

### Decision · Program_Chairs · 2025-01-22

Reject